# Autonomy or Working Conditions?—Research on Heterogeneity and Influencing Mechanism of Self-Employment on Job Satisfaction in China

**DOI:** 10.3390/ijerph20010282

**Published:** 2022-12-24

**Authors:** Yizhi Han, Jingyi Wang

**Affiliations:** 1School of Labor and Human Resources, Renmin University of China, Beijing 100872, China; 2Nanyang Center for Public Administration, Nanyang Technological University, Singapore 639798, Singapore

**Keywords:** informal employment, self-employment, job satisfaction level, autonomy, working condition, China

## Abstract

The development of globalization and information technology has been promoting informal work rapidly. In this process, self-employment is gradually becoming an important employment approach. As two of the key variables, self-employment laborers’ work autonomy and work conditions are largely discussed as the potential determinants of their work satisfaction. Which of these two factors is actually influencing labors’ satisfaction level? So far, relevant studies are insufficient to respond to this question, especially in developing countries. This study investigates the influence of work autonomy and working conditions on self-employed workers’ job satisfaction in China. China Labor-force Dynamics Survey data is used to examine the impact of self-employment on workers’ job satisfaction and the influence mechanism of work autonomy and working conditions. Propensity score matching and instrumental variable methods were applied to avoid sample selection bias and endogeneity. We found that self-employment has a significant negative effect on job satisfaction; poor working conditions are responsible for self-employed workers’ lower satisfaction level, and self-employment behavior impacts job satisfaction differently in terms of the type and gender of self-employed workers. Therefore, stronger social security and better working conditions for the self-employed should be provided.

## 1. Introduction

With globalization and the development of technology, especially the advancement of the Internet, although the ‘pushing and drawing’ effect of cost and technology has diversified the employment mode of companies, it has also strengthened the de-employment tendency [1,2]. With the popularization of informal employment [3] and development of urbanization in China [4], self-employment has emerged as an important employment mode [5,6]. According to the World Bank data, as of 2019, 44.7% of the employed population in China is self-employed. This phenomenon has been further strengthened by the promotion of a new economy, drawing research attention towards this group of workers. So far, research on job satisfaction has mainly focused on formal workers, with only a few studies on self-employed workers. Therefore, the mechanism of self-employed workers’ job satisfaction determinants is not conclusive.

There are mainly two accepted viewpoints regarding the influence mechanism of self-employed workers’ job satisfaction. The first is that, compared with formal employees, self-employed workers are more satisfied with their work [7,8] because they have higher job autonomy [5,9]. The second viewpoint is that, because of poor working conditions in general, self-employed workers’ job satisfaction is lower than that of formal employees [10,11]. Moreover, as self-employed workers are economically dependent on a few ‘customers’, they not only have limited autonomy [6,12] but are also not covered by the labor rights granted to employees in the traditional employment scenario [1,5,13]. A large number of empirical studies have proven that, compared with formal employees, self-employed workers endure longer working hours [11,14], have lower income, and receive inadequate social insurance protection [15]. The conflict of viewpoints mainly comes from the different research perspectives and the uncertain selection of research groups.

Based on the discussion above, three questions are raised as the research aims of this study:(1)How is the current job satisfaction level of self-employed workers in China compared with the employed?(2)What is the decisive factor (job autonomy or working conditions) that impacts self-employed workers’ job satisfaction level?(3)What is the influence mechanism of the decisive factor based on Chinese data?

A review of the existing studies on this topic reveals several research gaps. First, although it is widely admitted that the self-employed labor group is heterogeneous [6,16], which could influence analysis results [17], most of the current studies consider it a homogeneous group, thus barely examining different types of self-employed workers and gender differences while conducting research, and this leads to the difference of final results to a great extent. Meanwhile, the difference of the sample’s background and behavioral motivation will influence the final results a lot, and that’s why contradictory research results usually appear. In regard to research methods, there is a self-selection bias as a person himself/herself chooses to be self-employed or employed, which affects the validity of research results. However, many studies ignored the self-selection problem, and it of course caused the problem of estimation bias. In addition, there are problems of reverse causality between self-employment and job satisfaction, missing variables, and the endogeneity issue, which can lead to result bias. Therefore, this paper attempts to explore the role of autonomy and working conditions in influencing the job satisfaction of self-employed people and their role paths by distinguishing the groups of self-employed people.

To bridge these research gaps and replenish positivist proofs on this topic, we utilized data from the China Labor-force Dynamics Survey (CLDS) for analysis. The individual data was organized and collected by Sun Yat-sen University in 2016. To overcome sample self-selection and endogenous bias, propensity score matching (PSM) and instrumental variables methods were employed. The results of our research show that (1) self-employed workers’ satisfaction level is significantly lower than that of workers who have a formal job; (2) self-employed workers’ autonomy is not significantly higher than that of employees—thus, autonomy is not a mediator variable that affects self-employed workers’ job satisfaction; (3) working conditions have a mediating effect on self-employed workers’ job satisfaction; and (4) the job satisfaction of self-employed workers is heterogeneous in terms of the type and gender of workers.

## 2. Literature Review

The International Labor Organization (ILO) defines self-employment as an employment mode including employers and self-employed workers such as freelancers and independent contractors (who do not hire any employees for their business). Self-employed workers thus form a large and complex group [18], consisting of employers who own abundant human and social capital [1], as well as people belonging to social marginal groups who have no choice but be self-employed because of discrimination in the formal sector [19]. Because of the differences between self-employed workers’ human and social capital, their motivations and appeal for choosing self-employment work vary [20]. This heterogeneity makes it difficult for researchers to obtain unanimous results about the influence of self-employment behaviour on job satisfaction. As this study aims to investigate the impact of self-employment behaviour on workers’ job satisfaction, it is necessary to ensure that, besides the employment mode, the basic conditions between the experimental group (self-employed) and the reference group (employed) are the same. Therefore, the self-employed workers investigated in the present study are self-sustaining laborers from non-agricultural sectors; they work independently without employer(s) or employee(s) of their own.

### 2.1. Effect of Self-Employment on Job Satisfaction

There is no consensus among researchers on self-employed workers’ job satisfaction. Scholars who are more optimistic believe that job satisfaction of self-employed workers is higher than that of employees as they can decide their working hours, workplace, work manner, and other working conditions; thus, they are able to balance work and family [8]. However, studies show that job satisfaction of self-employed workers is lower than that of formal employees [21]. This is because, on the one hand, they are low-skilled workers with a weak position in the labor market who are forced to be self-employed in order to survive [22]. On the other hand, a majority of self-employed workers are dependent and covert employees who are not legally employed by the employers [23]. Their income depends on one or a few clients. Thus, not only can they not enjoy the labor rights of formal employees, but they also do not have the advantage of the autonomy of self-employed workers for independence; at the same time, they are responsible for the work results. Further, it is difficult for them to develop a small business out of self-employed work without the starting capital that is necessary for the production of additional profits and capital accumulation. Based on the above analysis, we propose the following hypothesis.

The conflicting results of these studies show the complexity of job satisfaction among the self-employed. First of all, the backgrounds of the analysis samples will affect the final results of the research. Workers in developed countries and developing countries are likely to choose self-employment because of different motivations, and self-employed people in countries with different economic level also have certain differences in labor skills [24]. Secondly, studies conducted from different perspectives may also achieve different results. Demand-driven analysis points out that the informal employment is caused by limited job quantity and unsatisfactory job quality while the motivation-driven view holds that workers’ choice of informal occupation is the embodiment of self-utility maximization choice, which reflects their preference. The discussion of various trends leads to different conclusions [25]. Finally, there is heterogeneity within the self-employed, and different self-employed people are faced with different constraints and demand preferences. Therefore, studies on different types of self-employed people are likely to show diverse results [26].

Based on the above discussion and preliminary analysis of China’s national conditions, we assume:

**H1.** *Self-employed workers have lower job satisfaction compared with employed workers*.

### 2.2. Effect of Job Autonomy on Self-Employed Workers’ Job Satisfaction

Autonomy refers to the ability of workers to control work-related affairs, such as working time, working style, and working content. Existing studies have proved that there is a positive correlation between autonomy and job satisfaction. Nevertheless, whether self-employed workers have autonomy is still debatable. Scholars with an optimistic attitude believe that autonomy is the main feature of self-employment and is the biggest motivation for workers, especially for female workers. In particular, some scholars who study platform employment believe that those working on an online platform have complete work autonomy. They can not only choose their working hours and working place, but also the platform they want to work on. However, others have revealed the gap between the ideal and reality of self-employment, which is called the ‘autonomy paradox’ [11]. Researchers argue that self-employed workers have no autonomy [1,5], they rely on one or two employers financially, and therefore have only limited autonomy. 

In line with the Informal Employment Report of ILO, the self-employed situation is regarded as ‘dependent’ or ‘concealment’ employment; furthermore, it is argued that this so called ‘autonomy’ results in the loss of rights of self-employed workers compared with traditional employees; thus, they have lower job satisfaction than workers in traditional employment relationships. The situation worsens when, under higher autonomy, independent contractors voluntarily choose to work longer hours (working overtime at night, on holidays) and do not participate in social security [6]. Moreover, there is ongoing debate about whether platform workers have autonomy. Some studies have shown that platform control and worker autonomy coexist in platform employment [27]; others believe that the platform has stronger and more covert control over workers through technological means [28,29]. Therefore, platform workers do not have job autonomy. Based on the above discussion, we propose the following hypothesis:

**H2.** *Work autonomy of self-employed workers is not higher than that of employed workers*.

### 2.3. Effect of Current Working Conditions on Self-Employed Workers’ Job Satisfaction

In this paper, the working conditions of the self-employed are defined as the comprehensive factors composed of working income, working hours, and job security. Studies have shown that the income of self-employed workers includes profits from investment and operation; therefore, their wages are higher than those of employed workers, which is another factor that attracts workers to engage in self-employment [30,31]. However, an increasing number of studies have found that self-employed workers earn less than employed workers [32]. As independent contractors in the labor market, freelancers are separated from each other in terms of time and space; thus, there is a competitive relationship between them, making it difficult for them to negotiate with their ‘employer’ through the right of solidarity. Further, there is virtually no barrier to entry in the self-employment market, resulting in an oversupply of labor. Therefore, clients have the monopoly to set the price, while self-employed workers can only choose to accept, otherwise losing the job opportunity [1]. Lastly, self-employed workers lack the guarantee of basic labor rights such as minimum wage. Consequently, self-employed workers earn less than employed workers in many situations, which reduces job satisfaction [33].

Several studies have confirmed that self-employed workers work longer hours than employed workers [6,34]. The reasons are as follows: first, because of the low wage, self-employed workers have a relatively strong desire to increase their income; thus, the substitution effect outweighs the income effect, and they increase their income by working longer hours. Further, unlike the time-based wage system of the formally employed, self-employed workers derive their income based on their work. For the self-employed, no work means no income. Therefore, in the absence of maximum hour limits, self-employed workers tend to work overtime. In addition, to ensure the continuity of income, self-employed workers need to invest time in self-improvement learning and maintaining customer relationships. Thus, they work significantly longer hours than formal employees, leading to decreased job satisfaction.

Because of the limitation of payment capacity and the institutional division of social insurance between urban and rural areas in China, the possibility of self-employed workers participating in social insurance is significantly lower than that of formally employed workers. Even if self-employed workers participate in social security, the coverage is significantly less than that of formally employed workers [35,36]. Such an institutional segregation not only hinders the identity of the self-employed [37], but also weakens their ability to resist risks, indicating that they are in greater need of social insurance. Therefore, we believe that absence of social security reduces job satisfaction of self-employed workers, and hypothesize the following (see Figure 1).

**H3a.** 
*Logarithm of income of self-employed workers is significantly lower than that of the employed, resulting in a decrease in job satisfaction.*


**H3b.** 
*Working hours of self-employed workers are significantly higher than those of the employed, leading to a decrease in job satisfaction.*


**H3c.** 
*Self-employed workers have less social security than employed workers, resulting in lower job satisfaction.*


**H3d.** 
*Working conditions are the intermediary mechanism that affects job satisfaction of self-employed workers.*


### 2.4. Heterogeneity within Self-Employed Workers

The formation of self-employment theory shows that the choice of self-employment is influenced by both active and passive factors, and different motivations of self-employment will lead to differences in job satisfaction. Based on the heterogeneity of self-employed people and the analysis of job stability preference [38], and combined with the reasons why workers choose to be self-employed, this study divides self-employed workers into three categories: active start-up self-employed, passive start-up self-employed, and non-entrepreneurial self-employed. 

Active start-up self-employed workers are the workers who choose to be self-employed although they have good job opportunities, and the employment behavior belongs to active self-employment. Passive start-up self-employed workers are defined as those who choose to be entrepreneurs in the absence of good job opportunities. Non-entrepreneurial, self-employed workers refer to those who cannot get jobs in the market and do not have the capital to invest in entrepreneurship, including freelancers, part-time workers, street vendors, nannies without dispatching units, self-employed drivers, artisans, and so on. What needs to be distinguished is the difference between the passive entrepreneurial self-employed and the non-entrepreneurial self-employed. Passive start-up self-employed workers are better than the non-entrepreneurial self-employed in terms of social capital and property status, although they both have no good job opportunities and belong to the scope of passive self-employed.

Active start-up self-employed workers have abundant human and social capital [37]. They attach more importance to work autonomy [31] and have a high income [39], and human capital can be given full play [20]. Therefore, they choose to start their own businesses and act as their own ‘boss’. Thus, we propose the following hypothesis.

**H4a1.** 
*Compared with employed workers, self-employed workers who actively start their own businesses have higher job satisfaction.*


**H4a2.** 
*Work autonomy is the influencing mechanism that improves their job satisfaction (see Figure 2).*


Passive start-up self-employed workers engage in self-employment activities because of a lack of good job offers. Compared with employed workers, their competitiveness and income level are low. Therefore, they have to work longer hours to balance the input and output and to further ensure that their income is sustainable and develops [1]. However, spending more time at work not only affects workers’ health but also renders them unable to take care of their families [6]; all these factors undermine workers’ satisfaction with their jobs. In addition, such workers lack social security; therefore, when encountered with inevitable risks such as old age and illness, they are not only uninsured but also face the dilemma of losing their income due to a lack of labor output. Therefore, we propose the following hypothesis.

**H4b1.** 
*Compared with employed workers, passive start-up self-employed workers have lower job satisfaction.*


**H4b2.** 
*Working conditions are the intermediary mechanism affecting their job satisfaction (see Figure 2).*


With regard to the non-entrepreneurial self-employed, because of the limitations of human and social capital, they are not only unable to obtain employment in the formal labor market but also have no capital to invest to ‘be the boss’; therefore, they become ‘casual workers’ providing labor services and are socially marginalized. Most of them are employed in the informal sector, with low income, unstable jobs, poor income sustainability, long working hours, high turnover rates, no benefits, no job security or protection from state labor laws, and a lack of social security [21,22]. Thus, we propose the following hypothesis.

**H4c1.** 
*Non-entrepreneurial self-employed workers have lower job satisfaction compared with employed workers.*


**H4c2.** 
*Working conditions are the intermediary mechanism that affects their job satisfaction (see Figure 2).*


Some scholars believe that for female self-employed workers, self-employment can balance the relationship between family and work [11,18]. In addition, self-employment can free female workers from gender discrimination and career bottlenecks prevalent in the formal sector. For these reasons, the negative impact of self-employment on job satisfaction is relatively less among female self-employed workers. Thus, we propose the following hypothesis.

**H5.** 
*Job satisfaction of female self-employed workers is higher than that of male self-employed worker (see Figure 1).*


To sum up, this paper puts forward the following hypothesis(see Table 1).

## 3. Materials and Methods

### 3.1. Data Sources

This study utilized data collected through CLDS, a survey conducted by the Social Science Research Center of Sun Yat-sen University in 2016. The survey conducted a biennial tracking survey of urban and rural residents in China and established a comprehensive database based on the sample of labor force, including the tracking and cross-sectional data of individual, family, and community. With its extensive and reliable survey, it provides high-quality basic data for empirically oriented theoretical research and policy research. It is one of the most important data sources for the study of China’s labor market now.

CLDS used a multi-stage, multi-level, probability sampling method proportional to the size of the labor force. It takes the lead in the rotation sample tracking method in China, which can not only better adapt to the drastic changing environment in China, but also pay attention to the characteristics of cross-sectional survey. The sampling design was to randomly divide the samples into 4 samples. Each sample was followed for 4 consecutive rounds (6 years) and then the survey was quit while a new rotating sample was used to supplement. The sample included working people aged 15–64 years as the research subjects. The survey focused on the current situation and changes in education, employment, labor rights, professional satisfaction, and happiness of the labor force. The survey covered 29 provinces and cities in China with a sample size of 401 villages, 14,226 households, and 21,086 individuals. Thus, the data represents the whole nation: eastern district, central district, and western regions of China; in addition, it also covered Guangdong Province and the Pearl River Delta of China.

The questionnaire was divided into nine parts: worker background, education experience and migration history, job-related information, work history, entrepreneurial process, social participation and support, worker status, reproduction, and health status. In the form of a household survey, questionnaire survey is combined with an interview survey, and the quality of the collected data is actively guaranteed by strictly controlling the investigators, the influence of the visitors, and the variables of the investigation environment and time. The data disclosed in 2016 is a relatively complete and reliable data developed on the basis of the survey in 2014 by rotating sample tracking. The validity and reliability of the questionnaire and survey data have also been verified in the follow-up discussion [40,41].

In consideration of the regulation of employment age in China, we chose people aged 16 years and above as the research subject, obtaining a sample of 7105 workers after excluding employers, farmers, and unemployed workers. According to the four basic principles that delivered by Roscoe in 1975 [42], the sample size must be ten times or more than the number of variables. Our study takes 14 variables into account, including independent variables, dependent variables, mediating variables, and control variables, so the sample of 7105 is suitable for our study.

### 3.2. Description of Variables

The independent variables included self-employed, active start-up self-employed, passive start-up self-employed, and non-entrepreneurial self-employed. All independent variables were dummy variables and were assigned the value 0. The value for the variable self-employed was obtained by confirming the job type of interviewees through a survey questionnaire. After excluding the samples of employers and agricultural workers, self-employed workers were assigned the value 1.

To distinguish among active start-up self-employed, passive start-up self-employed, and non-entrepreneurial self-employed, this study further focused on the question ‘Did you start your business because you found a good entrepreneurial opportunity or because you had no better job options?’ Active self-employed workers were those who chose ‘seize good entrepreneurial opportunities’ and ‘there are good jobs at that time, but the entrepreneurial opportunities are better’, and were assigned the value 1; passive self-employed workers chose ‘no better job options’ and ‘both good entrepreneurial opportunities and no better job options’, and were assigned the value 1. The remaining workers were placed under the non-entrepreneurial self-employed category and were assigned the value 1.

Job satisfaction was the dependent variable and was measured through 11 questions based on ‘income’, ‘job security’, ‘work environment’, ‘working hours’, ‘promotion’, ‘interesting work’, ‘working partner’, ‘ability and skills to use’, ‘respect from others at work’, ‘opportunity of expressing their views at work’, and ‘overall satisfaction with work’. The questions covered all aspects of the job satisfaction, and the answers ranged from 1 = very satisfied, 2 = satisfied, 3 = general, 4 = not satisfied, to 5 = very dissatisfied. 

To better explain the relationship between these variables, this study reassigned and reverse coded the options, defining the missing value as ‘uncertain’ and assigning it to 1. Exploratory factor analysis was performed to reduce these 11 dimensions to a single common factor with an eigenvalue greater than 1 named as job satisfaction. The variance contribution rate of this factor was up to 95%, and the KMO value was greater than 0.8, which was suitable for factor analysis.

Based on the existing literature, we chose overtime work, logarithm of income, and social insurance participation as the mediating variables. The concept of working hours and overtime under China’s labor law applies only to employees while the working hours of self-employed workers are not regulated by law. In contrast to employees, taking into account the flexibility of working hours of self-employed workers, ‘overtime’ is considered a dummy variable, with 0 indicating non-delayed work and 1 referring to delayed work. Overtime is calculated as the self-employed worker’s ‘average number of working hours per day’ multiplied by 7 days minus the number of ‘days off per week’ greater than 21.75 days multiplied by 8 h.

The logarithm of income is a continuous variable and is measured by the logarithm of the total income of the respondents in 2015 (Although there are questions in the questionnaire about the interviewees’ salary income or business income in 2015, the authors found that some self-employed and employed workers earned both types of income. Therefore, this study sets total income as a proxy variable. In addition, there is the phenomenon of zero income in individual samples. Given that the workers included in the study sample had a job in 2015, it is against common sense for them to have an income of zero. Therefore, this study assigns the logarithmic average of corresponding wages according to the type of work to the samples with zero income). Social insurance participation is a dummy variable that includes answers to a series of ‘self-answering’ questions: ‘are you enrolled in pension, medical, (According to the provision in the “Social Insurance Law” of China that persons in flexible employment may participate in the basic endowment and medical insurance, and considering the differences in benefits of endowment and medical insurance among different groups, this study investigates whether the interviewees participate in the endowment and medical insurance by checking whether they participate in the urban employees’ endowment and medical insurance.) work injury, maternity, and unemployment insurance?’ As long as the respondents participate in one type of insurance, we code it as ‘1’; otherwise, the variable is coded ‘0’. To evaluate his/her work autonomy, interviewees respond to three questions concerning ‘work task content’, ‘work schedule arrangement’, and ‘workload/work intensity’. Work autonomy was obtained by extracting common factors through reverse coding and exploratory factor analysis. The KMO value was greater than 0.8, which is suitable for factor analysis.

The control variables included personal characteristics (age, gender, education, marital status, number of children), nature of hukou (agricultural hukou), and location of hukou (migration). Based on available studies, we believe that these factors have a potential impact on workers’ job satisfaction. Considering the differences in social, economic, and cultural environment of different regions in China, this study added province as a control variable to control the unobservable factors related to regions that do not change over time [31]. In addition, considering the non-linear relationship between age and job satisfaction, the square of age was added as another control variable. Finally, because of differences in self-employment behaviors among different industries, industry is added to the model as a control variable, and we use different R&D investments to measure industry differences. 

Among them, gender, marital status, province, household registration nature, and household registration location are all assigned values as dummy variables. The number and age of children are measured according to the number of survey results. Educational attainment is divided into four grades (elementary and below, middle school, high school, college and above) with a rating of 1 to 4 from lowest to highest. The difference between industries is measured by the difference in the number of enterprises with annual R&D activities within the industry. The descriptive statistics of core variables are shown in Table 2.

### 3.3. Research Framework and Methods

This study adopts OLS and logit models to explore the effect of self-employment on workers’ job satisfaction and its influencing mechanism. Referring to existing research on why people choose to engage in self-employment [20] and as gender has an impact on job satisfaction [6,7,43]. This study adopts gender as a moderating variable to analyze the heterogeneity of the effect of self-employment on job satisfaction.

As there may be missing variables and inverse causality between the independent variable self-employment and the dependent variable job satisfaction, this study adopts instrumental variables and two-stage least squares (2SLS) estimation methods to overcome the estimation bias of the dependent variable caused by endogeneity.

This study uses ‘number of enterprises with R&D input in 2011’ as the instrumental variable. This is because technological progress can improve the operational efficiency of organizations and labor production efficiency, so that a large amount of ‘surplus’ labor is released into the labor market, which is more likely to be self-employed. On the other hand, technological innovation will promote organizational change and encourage enterprises to ‘de-employ’ their internal secondary departments and secondary jobs in the form of outsourcing, producing a number of work opportunities for the self-employed [44]. Thus, technological progress leads, either positively or negatively, to more self-employment. However, enterprises’ investment in research and development (R&D) activities will not lead to immediate improvement in their overall performance, nor will it lead to rapid organizational changes. It will take years for companies to start investing in R&D to increase the number of self-employed opportunities. After testing, the instrumental variable ‘number of enterprises with R&D input in 2011’ was found to be significantly correlated with self-employment. The F value of regression in the first stage is 14.39, excluding the possibility of weak instrumental variables. The Hausman test results are also significant, indicating that the independent variable self-employed is endogenous and suitable for the instrumental variables method.

Considering the self-selection of self-employed workers, although more control variables were added to the model, this study could not effectively reduce the estimation bias of the effect of self-employed behavior on job satisfaction. Therefore, this study adopted the PSM method to control the self-selective bias of samples. The Propensity Score Matching method is to find every self-employed person who has the same or similar characteristics in other aspects (covariates) except for different employment choices through propensity score matching, and take their job satisfaction as the “counterfactual” job satisfaction of the individual samples of the processing group, so as to maximize the estimation error caused by sample selectivity bias [45]. Referring to existing literature, this study includes age [7], gender [6,7,43], marital status [43,46], agricultural Hukou [47], migration status, education experience, number of children, province, and industry as covariables influencing the employment choice of workers. To ensure the robustness of the results, radius and kernel matching methods based on different matching principles were used to match the samples. Before matching, we tested the sample balance of covariables in the treatment and control groups. The test results show that there was significant selectivity bias before matching, which became insignificant after matching, indicating that the above matching methods effectively solved the problem of sample selectivity bias. The sample size after radius matching comprised 6600 workers, 5098 of whom were in the control group and 1502 in the experimental group. The average treatment effect (ATT) was −0.29 and was significant at the 1% level. After completing tendency score matching, the samples were respectively added to the OLS or logit model to analyze the influence of self-employment on job satisfaction.

Outliers in this study were determined according to the 3σ principle and data outliers were regarded as missing values, which were processed together with missing data. In terms of processing missing data, since the original data in this study was relatively complete with few missing values, regression could be adopted to complete the missing values. The regression equation (model) was established based on the complete part of the data set. Then, the known attribute value was substituted into the equation to estimate the unknown attribute value for the object containing null value, and the estimated value was used to fill the vacancy. Finally the analysis sample with complete data was obtained.

## 4. Results

The empirical results focus on four aspects: (1) impact of self-employment on workers’ job satisfaction; (2) whether autonomy is an intermediary mechanism that affects job satisfaction of self-employed workers; (3) whether working conditions are an intermediary mechanism affecting job satisfaction; and (4) whether gender and the reasons for choosing to be self-employed affect the job satisfaction of self-employed workers.

### 4.1. Effect of Self-Employment on Job Satisfaction

#### 4.1.1. Basic Results

Table 3 presents the basic findings of this study. Columns (1) to (3) show the regression results after the addition of control and adjustment variables based on the linear probability model.

As can be seen in column (1), the job satisfaction of self-employed workers is significantly lower than that of employed workers. After adding the control variables province and industry to the model, job satisfaction of self-employed workers is still negative; however, the level of job dissatisfaction has significantly reduced. The results in columns (1) and (2) show that self-employment does reduce workers’ job satisfaction, verifying hypothesis H1. After adding gender as the adjustment variable in column (3), although the coefficient increases, the increase is not significant. This might have been caused by endogeneity; therefore, 2SLS estimation method is used to resolve the endogenous errors. From the results of column (4), it can be seen that even after the addition of instrumental variables, job satisfaction of self-employed workers is still significantly low, verifying the robustness of hypothesis H1. Column (5) shows the results of the moderating effect of gender. We can see that the interaction term is significant and positive, indicating that, compared with men, self-employment has a smaller inhibitory effect on women’s job satisfaction. This shows that gender plays a positive role in adjusting the job satisfaction of self-employed workers; that is, it alleviates the inhibitory effect of self-employment on job satisfaction, thus verifying hypothesis H5.

#### 4.1.2. Heterogeneity of Job Satisfaction within the Self-Employed Group

Considering the internal heterogeneity of motivations (active or passive) and types (entrepreneurial or non-entrepreneurial) of self-employed workers in choosing to be self-employed, leading to differences in job satisfaction, Table 4 is divided into six columns. Columns (1), (3), and (5) show the influence of active self-employment, passive self-employment, and non-entrepreneurial self-employment on job satisfaction in the absence of control variables province and industry. Columns (2), (4), and (6) present the estimated results after the addition of control and dummy variables.

A comparison of the results in columns (1) and (2) shows that, although active self-employment has a positive effect on job satisfaction, the effect is not statistically significant. Thus, hypothesis H4a1 is not verified. In column (3), the job satisfaction of passive self-employed workers decreased by 0.46 compared with that of employed workers. After adding the variables province and industry, although job satisfaction increased somewhat, it was still significantly negative, verifying the robustness of the negative influence of passive self-employment on job satisfaction. Column (5) shows that non-entrepreneurial self-employment significantly reduces workers’ job satisfaction. After adding control variables, the estimates increased and the negative impact of non-entrepreneurial self-employment on job satisfaction decreased. In addition, by comparing columns (4) and (6), we found that non-entrepreneurial self-employed workers have lower job satisfaction compared with passive self-employed workers.

### 4.2. Analysis of Influence Mechanism of Self-Employed Workers’ Job Satisfaction

#### 4.2.1. Autonomy

Table 5 shows the overall effect of work autonomy as an intermediary mechanism on self-employed workers’ job satisfaction. We can see that the effect of self-employment on job satisfaction in the first stage is significantly negative, with a coefficient of 0.28. When self-employment regresses to autonomy, self-employed workers are found to have higher work autonomy than employed workers. However, regression results of the second stage show that autonomy does not improve workers’ job satisfaction; rather, it further reduces it. In addition, the results of the second stage after the addition of autonomy are significant, and the absolute value is larger than that of the first stage, indicating that autonomy is not an intermediary mechanism affecting the job satisfaction of self-employed workers.

As for the mediated effect of autonomy on active self-employed workers’ job satisfaction, first-stage regression results of the job satisfaction of active self-employed workers are positive and not significant; however, the results of the second stage after the addition of autonomy are negative and significant, indicating that, although autonomy of self-employed workers is significantly higher than that of employed workers, autonomy does not bring higher job satisfaction to the workers. Thus, H4a2 is not verified.

Considering that problems such as endogeneity may lead to insignificant results, we carried out a robustness test to verify the results of hypothesis H2 using instrumental variables. Table 6 shows that, after replacing potential endogenous variables with instrumental variables, the absolute value of regression results in the second stage is greater than that in the first stage. In addition, the data shows that the effect of self-employment on autonomy became insignificant, indicating that work autonomy of self-employed workers was not significantly higher than that of employed workers. Thus, hypothesis H2 was assumed to be verified.

#### 4.2.2. Working Conditions

##### Overtime

Table 7 shows the overall effect of self-employment on job satisfaction mediated by overtime. We can see that the effect of self-employment on job satisfaction in the first stage is significantly negative with a coefficient of 0.28. As self-employed workers prefer working overtime than employed workers do, we consider overtime as a factor in the second-stage regression. After adding this variable, the regression results are significant, and the absolute value is smaller than that in the first stage. This indicates that working overtime is a partial intermediary mechanism that affects the job satisfaction of self-employed workers, thus verifying hypothesis H3b. 

Regarding the effect of passive self-employment and non-entrepreneurial self-employment on job satisfaction mediated by overtime, both types of self-employed groups are more likely to work overtime than employed workers. In addition, regression results of job satisfaction in the first stage are significantly negative, and the results of the second stage after the addition of overtime are significant, and the absolute value is less than the results in the first stage. This proves that overtime is an intermediary mechanism affecting job satisfaction of passive self-employed and non-entrepreneurial self-employed workers.

##### Social Security

Table 8 shows the overall effect of self-employment on job satisfaction mediated by social security participation. We see that the effect of self-employment on job satisfaction in the first stage is significantly negative with a coefficient of 0.28. Considering that self-employed workers are more likely to lack social security than employed workers, results of the second-stage regression show that lack of social security significantly reduces workers’ job satisfaction. In addition, after taking social insurance into account, second-stage results are significant and smaller in absolute value than the first-stage results. This indicates that social insurance is a partial intermediary mechanism affecting job satisfaction of self-employed workers, thus verifying hypothesis H3c.

Regarding the effect of passive self-employment and non-entrepreneurial self-employment on job satisfaction mediated by social insurance, both types of self-employed groups are more likely to lack social security compared with employed workers; further, regression results of job satisfaction in the first stage are significantly negative. After adding overtime, the second-stage results are significant, and the absolute value is less than the first-stage results. This proves that social insurance participation is an intermediary mechanism that affects job satisfaction of passive self-employed and non-entrepreneurial self-employed workers.

##### Logarithm of Income

Table 9 shows the overall influence of self-employment on job satisfaction mediated by income. We see that the impact of self-employment on job satisfaction in the first stage is significantly negative with a coefficient of 0.28. Through the regression of the logarithm of income, it is found that the income of self-employed workers is significantly higher than that of employed workers. However, after adding the logarithm of income and self-employment factors in the regression equation of the second stage, we found that the augmentation of logarithm of income did not improve the job satisfaction of self-employed workers. Moreover, the absolute value is bigger than the first-stage results, indicating that the logarithm of income does not affect self-employed workers’ job satisfaction.

Regarding the effect of income on the job satisfaction of passive self-employed and non-entrepreneurial self-employed workers, the absolute value of the regression result of job satisfaction of passive self-employed workers in the first stage is less than that in the second stage. Therefore, for passive self-employed workers, the logarithm of income is not an intermediary factor influencing job satisfaction; for non-entrepreneurial self-employed workers, however, the fact that their income is significantly lower than that of employed workers leads to a decrease in job satisfaction. In addition, the second-stage regression results are significant, and the absolute value is less than the first-stage results, indicating that the logarithm of income is a partial intermediary mechanism influencing the job satisfaction of non-entrepreneurial self-employed workers.

To test whether the OLS method has a bias in the estimated results of job satisfaction of self-employed and passive self-employed workers, we used 2SLS regression estimation, applying ‘number of enterprises with R&D input in 2011’ as the instrumental variable. The regression results are shown in Table 10.

Table 10 shows that logarithm of income of self-employed and passive self-employed workers is significantly lower than that of employed workers, indicating that the results of OLS estimation are biased because of endogeneity. After adding the instrumental variables, the equation coefficients of job satisfaction in the two stages for self-employed and passive self-employed workers are significant, and the absolute value is less than that in the first stage. This proves that logarithm of income is a partial intermediary mechanism affecting job satisfaction of self-employed and passive self-employed workers, thus verifying hypothesis H3a.

Based on the results shown in Table 7, Table 8, Table 9 and Table 10, hypotheses H4b1, H4b2, H4c1, and H4c2 are also verified. To conclude, of the thirteen hypotheses proposed in this study, eleven could be proved while two could not be verified. Figure 3 presents the influence mechanism of self-employed workers’ job satisfaction.

## 5. Discussion

Since the 19th Communist Party of China National Congress, the Chinese government has gradually shifted its concern from the quantity and structure of employment to the quality of employment. As an important indicator to measure the quality of employment, research on job satisfaction mainly focuses on employed workers while studies about self-employed workers are still insufficient. There is no consensus, either at home or abroad, on whether self-employment improves job satisfaction. While some believe that self-employed workers have higher job autonomy than employed workers do, and hence, their job satisfaction should be higher, others argue that the working conditions of self-employed workers are worse than those of employed workers, which reduces workers’ job satisfaction.

In China, is the effect of self-employment on workers’ job satisfaction positive or negative? How does autonomy or working conditions affect this mechanism? Do the reasons behind choosing to be self-employed affect workers’ job satisfaction, thus introducing heterogeneity? To answer these questions, this study takes self-employed workers as the research object and employed workers as the reference object to clarify how self-employment behaviour influences workers’ job satisfaction and the mechanism underlying this process. Existing studies were found to have research methodology-related problems such as self-selection of samples and endogeneity, which affect the reliability and validity of empirical results.

Individual data collected through CLDS, a survey organized and implemented by Sun Yat-sen University in 2016, were utilized for analysis. OLS and logit models were used to investigate the effect of self-employment on job satisfaction and explore whether autonomy or working conditions affect job satisfaction of self-employed workers. It was found that self-employment behaviour has a negative influence on workers’ job satisfaction. This shows that, in China, job satisfaction of self-employed workers is significantly lower than that of employed workers. Further, poor working conditions of self-employed workers, such as longer working hours, less social security, and lower income, are found to mediate their job satisfaction, thus conforming with the viewpoint that poor working conditions can reduce self-employed workers’ job satisfaction [5]. We used OLS and 2SLS models to verify that autonomy is not an intermediary mechanism affecting the job satisfaction of self-employed workers. Though self-employment and autonomy were found to be positively correlated, the correlation was not statistically significant. This verifies the viewpoints of ‘limited autonomy’ [1] and ‘autonomy paradox’ [11], which argue that autonomy does not bring higher job satisfaction to self-employed workers. Considering their internal heterogeneity, this study divides self-employed workers into three categories (active start-up self-employed, passive start-up self-employed, and non-entrepreneurial self-employed) and discusses differences between the three types of self-employed workers and traditional employees in terms of job satisfaction and its influencing factors. Job satisfaction of passive self-employed and non-entrepreneurial self-employed workers was found to be consistent with that of self-employed workers. However, the effect of self-employment on job satisfaction was positive but not significant. Therefore, the CLDS data failed to verify hypothesis H4a1 and H4a2, which needs to be verified by subsequent studies. Furthermore, it was found that compared with male self-employed workers, the benefits of self-employment for female workers could moderate the negative effect of self-employment on job satisfaction.

Based on the survey data in China, it can be found that the proportion of self-employed people in the survey is small, which is one of the reasons why the self-employed behavior is inversely proportional to job satisfaction in China’s data. For the majority of passive and non-entrepreneurial self-employed people, poor working conditions lead to their dissatisfaction with self-employed work. Maslow’s hierarchy of needs theory states that only when basic needs are satisfied can high-level needs come into play. In addition, the hierarchy of human needs is closely related to a country’s level of economic, cultural, educational, scientific, and technological development. As China is a developing country, being in an employment relationship is a prerequisite for workers in China to enjoy labor rights, such as fixed working hours, income security, social insurance, and other benefits. Consequently, self-employed workers remain out of the social security system. In the absence of minimum wage, maximum hours, labor safety and health conditions, and social insurance protection, the higher level of need–autonomy cannot play a role in the job satisfaction of the self-employed. Because of the relatively terrible working conditions of self-employed workers, approaches for protecting this group are being explored worldwide. 

Compared with employees in the traditional employment relationship, job satisfaction of passive start-up self-employed workers is significantly lower. Workers in this group choose self-employment through entrepreneurship because there are no good job opportunities. They are weaker than active self-employed workers in terms of human capital but are better than non-entrepreneurial self-employed workers in respect of social capital and property status. They have a degree of capital independence, but their income depends on few ‘customers’ [23]. On the basis of these causes, we believe that such workers are in a subordination relationship and are self-controlled labors [48] or independent workers with work autonomy [49]. Therefore, they could be classified as ‘third class workers’ or ‘independent workers’ [50] and be given partial preferential protection under German and Japanese models. Furthermore, the focus of protection should be on improving the coverage of social insurance. On the one hand, focusing on the ‘big government’ perspective of social security, government should increase its input in social insurance premiums and the proportion of government expenditure on social insurance at all levels. Further, the willingness of self-employed workers to participate in insurance should be enhanced. On the other hand, the social insurance system should adapt to the new trend of changes in labor relations and use Internet technology to establish a social insurance technology platform that focuses on individual social security accounts and integrates all-round functions such as real-time monitoring of work and business [51]. In addition, in such a system, the ‘employer’ should assume corresponding social insurance liabilities according to the workload of the self-employed so that the social security payment and labor relations ‘unbind’ to achieve and improve the rate of self-employed participation.

In the CLDS database, the self-employed who did not engage in a start-up business were mainly rural migrant workers employed in construction, wholesale, retail, and catering industries. Because of their limited human and social capital, it is difficult for such workers to secure positions in the formal labor market. Therefore, self-employment has become their only choice for surviving [52]. However, although they are in a venerable position in the labor market, they are excluded from legal protection because of institutional factors. The lack of labor rights protection leads them to have low and unstable income, long working hours, low bargaining power, and lack of social security, which can reduce their job satisfaction.

Therefore, we believe that the legal labor system should pay attention to the problems faced by non-start-up self-employed workers. Moreover, we propose expanding the scope of labor law protection, absorb this group into the labor law protection system, and provide them with full protection [53]. Furthermore, the low human capital of this group is also the direct cause of their weak position. We believe that provision of vocational skills training courses for self-employed workers, either by training institutions or by the government, could enhance their work skills and competitiveness inside the labor market, thus improving their situation gradually.

The outbreak of the COVID-19 has a great impact on China’s labor market. On one hand, the global spread of the epidemic has affected the economic demand, shrinking the employment demand of some enterprises. Non-entrepreneurial self-employed people who lack stability are the first to be damaged. Due to the low skill level and low employment threshold of these groups, there are fewer job opportunities during the epidemic. The working conditions of non-self-employed people at risk of long-term unemployment have been further worsened. The unprosperous economic situation will also have a negative impact on the active self-employed and passive self-employed. The downturn in the form of entrepreneurship and investment leads to a decrease in their preference for autonomy and an increase in their preference for stable employment, thus the negative impact of self-employment on job satisfaction is further deepened. On the other hand, considering the safety of people’s lives, China adheres to the strategy of dynamic zero clearance, and some areas still take relatively strict epidemic prevention and control measures. However, it brings a certain negative impact on the recovery of normal economic order in China. It is more difficult for the self-employed to return to work while their working space is compressed [54,55]. In general, the outbreak of COVID-19 strengthens the negative impact of self-employment on job satisfaction, weakens the autonomy preference of the self-employed, worsens the working conditions of the self-employed, and reinforces the conclusion of this paper.

## 6. Conclusions

The findings include the following four points. (1) According to Chinese data, self-employment can significantly reduce workers’ job satisfaction while job satisfaction of female self-employed workers is higher than that of male self-employed worker. (2) Although the autonomy of self-employed workers is significantly higher than that of employed workers, autonomy does not bring higher job satisfaction to the workers, so work autonomy isn’t the influencing mechanism that improves their job satisfaction. (3) Compared with employed workers, passive start-up self-employed workers and non-entrepreneurial self-employed workers have lower job satisfaction, and working conditions are the intermediary mechanism affecting their job satisfaction.

The theoretical contributions of this study are, hence, as follows. (1) This paper provides empirical proofs for the study of this topic from the Chinese context and enriches the data of developing countries in self-employment. (2) In terms of research ideas, by distinguishing the types of self-employed, this paper brings the demand-driven and motivation-driven self-employed into the same framework for discussion, so as to avoid conflicting research conclusions caused by different perspectives. (3) The study fully considers the influence of respondents’ background on their behavior choices and takes into account necessary factors as comprehensively as possible to ensure the integrity and reliability of the research results. (4) This study applies the PSM method and the 2SLS estimation method to control the self-selective bias of samples, which helps solve the problem of estimation bias caused by sample self-selection and endogeneity. This paper also provides some useful policy advice, which contributes to the improvement of China’s labor market.

The article has the following limitations as well. First of all, the gender factor will certainly have an impact on the job satisfaction of the self-employed. Since this paper focuses on the relationship between self-employment behavior and satisfaction, as well as the paths that different types of self-employed workers are affected by self-employment behavior, only a brief discussion is made on the gender factor. Scholars may not only consider further exploring the influence path of the gender factor on the job satisfaction of self-employed people, but also explore the differences in the influence of gender factors in various types of self-employed groups. Secondly, this paper uses Chinese data to investigate the labor mode of self-employment, and it remains to be explored whether the final results are scalable. Further studies can be carried out based on data from other developing or developed countries to compare the differences in results and explore the reasons behind them. Finally, a deeper study can be carried out on the impact of COVID-19 on the labor market of various countries. This paper only discusses whether COVID-19 will have a conflicting impact on the conclusion of this paper based on the facts of China, without conducting a quantitative in-depth study.

## Figures and Tables

**Figure 1 ijerph-20-00282-f001:**
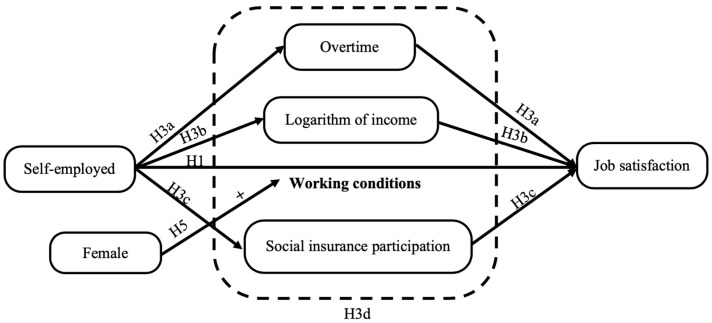
Model of the impact mechanism of self-employment on job satisfaction.

**Figure 2 ijerph-20-00282-f002:**
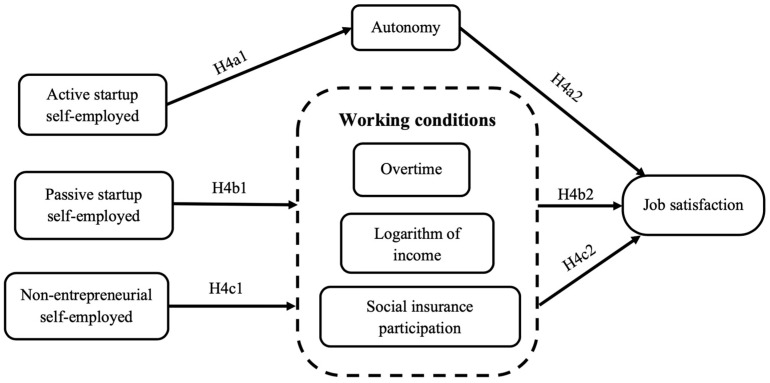
Mechanism model of classified self-employment affecting job satisfaction.

**Figure 3 ijerph-20-00282-f003:**
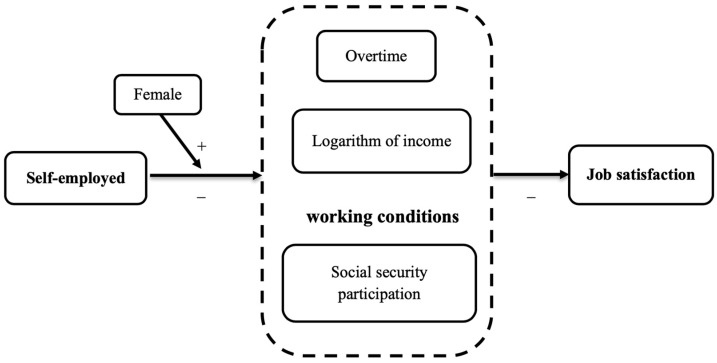
The impact of self-employment on job satisfaction and its mechanism.

**Table 1 ijerph-20-00282-t001:** All the hypotheses.

Serial Number		Hypothesis
H1		Self-employed workers have lower job satisfaction compared with employed workers.
H2		Work autonomy of self-employed workers is not higher than that of employed workers.
H3	H3a	Logarithm of income of self-employed workers is significantly lower than that of the employed, resulting in a decrease in job satisfaction.
	H3b	Working hours of self-employed workers are significantly higher than those of the employed, leading to a decrease in job satisfaction.
	H3c	Self-employed workers have less social security than employed workers, resulting in lower job satisfaction.
	H3d	Working conditions are the intermediary mechanism that affects job satisfaction of self-employed workers.
H4	H4a1	Compared with employed workers, self-employed workers who actively start their own business-es have higher job satisfaction.
	H4a2	Work autonomy is the influencing mechanism that improves their job satisfaction.
	H4b1	Compared with employed workers, passive start-up self-employed workers have lower job satisfaction.
	H4b2	Working conditions are the intermediary mechanism affecting their job satisfaction.
	H4c1	Non-entrepreneurial self-employed workers have lower job satisfaction compared with employed workers.
	H4c2	Working conditions are the intermediary mechanism that affects their job satisfaction.
H5		Job satisfaction of female self-employed workers is higher than that of male self-employed worker.

**Table 2 ijerph-20-00282-t002:** Descriptive statistics of core variables (Unit: person).

Variables	Value	Observations	Ratio	Mean	Standard Error	Min	Max
Self-employed	0 (employee)	5365	75.51%	0.24	0.43	0	1
	1 (self-employed)	1740	24.49%				
Active start-up self-employed (ASS)	0 (employee) 1 (ASS)	5365 325	94.29% 5.71%	0.06	0.23	0	1
Passive start-up self-employed (PSS)	0 (employee) 1 (PSS)	5365 846	86.38% 13.62%	0.14	0.34	0	1
Non-entrepreneurial self-employed (NES)	0 (employee) 1 (NES)	5365 569	90.41% 9.59%	0.10	0.29	0	1
Gender	0 (male)	4051	57.02%	0.43	0.50	0	1
	1 (female)	3054	42.98%				
Education experience	1 (elementary and below)	1302	18.33%	2.53	1.05	1	4
	2 (middle school)	2481	34.92%				
	3 (high school)	1555	21.89%				
	4 (University and above)	1767	24.87%				
Marital status	0 (married)	5962	83.91%	0.16	0.37	0	1
	1 (unmarried)	1143	16.09%				
Migration	0 (local)	5977	84.12%	0.16	0.37	0	1
	1 (migrant)	1128	15.88%				
Agricultural hukou	0 (non-agricultural)	2892	40.70%	0.59	0.49	0	1
	1 (agricultural)	4213	59.30%				
Social insurance participation	0 (no)	3542	49.85%	0.50	0.50	0	1
	1 (yes)	3563	50.15%				
Overtime	0 (no)	3971	55.89%	0.44	0.50	0	1
	1 (yes)	3134	44.11%				
Logarithm of income		7105		10.36	0.84	5.30	14.51
Autonomy		7105		2.45 × 10^−8^	0.95	−1.17	1.30
Job satisfaction		6886		−8.27 × 10^−10^	0.94	−4.09	1.27
Age		7105		41.31	11.42	16	84
Age squared		7105		1837.19	971.88	256	7056
Province		7105		40.10	13.33	11	65
Number of children		7105		0.44	0.77	0	6
Risk appetite		7105		3.33	0.71	1	5
Number of enterprises with R&D input in 2011		7105		2595.17	2483.22	23	8026

**Table 3 ijerph-20-00282-t003:** OLS and 2SLS regression results of the impact of self-employment on job satisfaction.

Variables	Job Satisfaction				
	(1)	(2)	(3)	(4) IV	(5) IV
Self-employed	−0.40 ***	−0.28 ***	−0.25 ***	−3.81 ***	−7.74 ***
	(0.03)	(0.03)	(0.03)	(1.13)	(3.68)
Female		0.08 **	0.09 **	−0.13	−1.53 *
		(0.04)	(0.04)	(0.09)	(0.80)
Self-employed × Female			−0.06		6.99 **
			(0.05)		(3.46)
Control variables	N	Y	Y	Y	Y
Province	N	Y	Y	Y	Y
Industry	N	Y	Y	Y	Y
observations	6600	6600	6600	6600	6600
$R^{2}$	0.03	0.08	0.08	−2.25	0.42

Notes: 1. The coefficient is a robust standardized regression coefficient; 2. The standard error in shown in parentheses; 3. *, **, and *** represent significant statistical levels at 10%, 5%, and 1%, respectively; 4. ”N” means that the variables are not included in the regression, while “Y” means that the variables are included in the regression.

**Table 4 ijerph-20-00282-t004:** OLS regression results of job satisfaction of different types of self-employed workers.

Variables	Job Satisfaction					
	(1)	(2)	(3)	(4)	(5)	(6)
Active start-up self-employed	0.01	0.08				
	(0.05)	(0.05)				
Passive start-up self-employed			−0.46 ***	−0.36 ***		
			(0.03)	(0.03)		
Non-entrepreneurial self-employed					−0.57 ***	−0.39 ***
					(0.05)	(0.05)
Control variables	N	Y	Y	Y	Y	Y
Province	N	Y	Y	Y	Y	Y
Industry	N	Y	Y	Y	Y	Y
Observations	5404	5404	5881	5881	5511	5511
$R^{2}$	0.08	0.06	0.03	0.08	0.03	0.08

Notes: 1. The coefficient is a robust standardized regression coefficient; 2. The standard error is shown in parentheses; 3. *** represents significant statistical levels at 1%; 4. ”N” means that the variables are not included in the regression, while “Y” means that the variables are included in the regression.

**Table 5 ijerph-20-00282-t005:** OLS regression results of the effect of self-employment on job satisfaction mediated by autonomy.

Variables	Autonomy	Job Satisfaction	
		First Stage	Second Stage
Self-employed	1.23 *** (0.03)	−0.28 *** (0.03)	−0.44 *** (0.03)
Active start-up self-employed	1.38 *** (0.05)	0.08 (0.05)	−0.11 ** (0.05)
Autonomy			0.14 *** (0.01)
Control variables	Y	Y	Y

Notes: 1. The coefficient is a robust standardized regression coefficient; 2. The standard error is shown in parentheses; 3, ** and *** represent significant statistical levels at 5% and 1%, respectively; 4, “Y” means that the variables are included in the regression.

**Table 6 ijerph-20-00282-t006:** 2SLS regression results of the effect of self-employment on job satisfaction mediated by autonomy.

Variables	Autonomy		Job Satisfaction			
			First Stage		Second Stage	
	(1)	(2) IV	(3)	(4) IV	(5)	(6) IV
Self-employed	1.23 *** (0.02)	0.231 (0.61)	−0.28 *** (0.03)	−3.81 *** (1.31)	−0.44 *** (0.03)	−4.02 *** (1.07)
Autonomy					0.13 *** (0.01)	0.913 *** (0.24)
Control variables	Y	Y	Y	Y	Y	Y
Observations	6600	6600	6600	6600	6600	6600

Notes: 1. The coefficient is a robust standardized regression coefficient; 2. The standard error is shown in parentheses; 3. *** represents significant statistical levels at 1%; 4. “Y” means that the variables are included in the regression.

**Table 7 ijerph-20-00282-t007:** OLS and logit regression results of self-employment’s impact on job satisfaction mediated by overtime.

Variables	Overtime	Job Satisfaction	
		First Stage	Second Stage
Self-employed	0.50 *** (0.01)	−0.28 *** (0.03)	−0.23 *** (0.03)
Passive start-up self-employed	0.55 *** (0.02)	−0.36 *** (0.03)	−0.30 *** (0.04)
Non-entrepreneurial self-employed	0.42 *** (0.02)	−0.39 *** (0.05)	−0.35 *** (0.05)
Overtime			−0.10 *** (0.02)
Control variables	Y	Y	Y

Notes: 1. The coefficient is a robust standardized regression coefficient; 2. The standard error is shown in parentheses; 3. *** represents significant statistical levels at 1%; 4. “Y” means that the variables are included in the regression.

**Table 8 ijerph-20-00282-t008:** OLS regression results of self-employment’s impact on job satisfaction mediated by social insurance.

Variables	Social Insurance Protection	Job Satisfaction	
		First Stage	Second Stage
Self-employed	−1.83 *** (0.09)	−0.28 *** (0.03)	−0.23 *** (0.03)
Passive start-up self-employed	−1.73 *** (0.11)	−0.36 *** (0.03)	−0.31 *** (0.04)
Non-entrepreneurial self-employed	−1.96 *** (0.18)	−0.39 *** (0.05)	−0.35 *** (0.05)
Social insurance protection			0.15 *** (0.03)
Control variables	Y	Y	Y

Notes: 1. The coefficient is a robust standardized regression coefficient; 2. The standard error is shown in parentheses; 3. *** represents significant statistical levels at 1%; 4. “Y” means that the variables are included in the regression.

**Table 9 ijerph-20-00282-t009:** OLS regression results of the effect of logarithm of income on job satisfaction.

Variables	Logarithm of Income	Job Satisfaction	
		First Stage	Second Stage
Self-employed	0.06 *** (0.02)	−0.28 *** (0.03)	−0.29 *** (0.03)
Passive start-up self-employed	0.05 *** (0.03)	−0.36 *** (0.03)	−0.37 *** (0.03)
Non-entrepreneurial self-employed	−1.17 *** (0.04)	−0.39 *** (0.05)	−0.36 *** (0.05)
Logarithm of income			0.18 *** (0.02)
Control variables	Y	Y	Y

Notes: 1. The coefficient is a robust standardized regression coefficient; 2. The standard error is shown in parentheses; 3. *** represents significant statistical levels at 1%; 4. “Y” means that the variables are included in the regression.

**Table 10 ijerph-20-00282-t010:** 2SLS regression results of the effect of logarithm of income on job satisfaction for self-employed and passive entrepreneurial self-employed workers.

Variables	Logarithm of Income		Job Satisfaction			
			First Stage		Second Stage	
	(1)	(2) IV	(3)	(4) IV	(5)	(6) IV
Self-employed	0.06 *** (0.02)	−6.83 *** (1.96)	−0.28 *** (0.03)	−3.81 *** (1.31)	−0.29 *** (0.03)	−2.37 *** (0.72)
Passive start-up self-employed	0.05 *** (0.03)	−6.07 *** (1.44)	−0.36 *** (0.03)	−3.61 *** (0.91)	−0.37 *** (0.03)	−2.49 *** (0.68)
Logarithm of income					0.18 *** (0.02)	0.21 *** (0.02)
Control variables	Y	Y	Y	Y	Y	Y
Observations	6600	6600	6600	6600	6600	6600

Notes: 1. The coefficient is a robust standardized regression coefficient; 2. The standard error is shown in parentheses; 3. *** represents significant statistical levels at 1%; 4. “Y” means that the variables are included in the regression.

## Data Availability

Individual data collected through CLDS, a survey organized and implemented by Sun Yat-sen University in 2016, were utilized for analysis.

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
