# Peer review of "Autonomy or Working Conditions?—Research on Heterogeneity and Influencing Mechanism of Self-Employment on Job Satisfaction in China"

_ijerph, 2022, doi:10.3390/ijerph20010282_

Round 1
Reviewer 1 Report
The manuscript presents a current topic of broad interest.
The information is properly organized, in a logical sequence, and exhibits strong documentation and data analysis. The data presented support the conclusions, and the bibliographic sources are properly cited.
I appreciate the article as interesting, well-documented, and meticulously and correctly elaborated; consequently, I suggest its publication in its existing form, without further revisions.

Author Response
Thank you very much for your encouraging comment!
Reviewer 2 Report
The main objective of this study is to find out effective factors include self-employment, … on job satisfaction . The topic is interesting, however, need some improvement.
1. In the title there are two “-“ which is not clear for me.
2. Title is too general, it would be better more specific.
3. First research question is too simple “What is the current job satisfaction level of self-employed workers in China?” not suitable.
4. End of introduction should be explain research gaps based on previous studies.
5. In introduction section (lines 70-76) authors mentioned that
6. “Results show that (1) self-employed workers’ 70 satisfaction level is significantly lower than that of workers who have a formal job; (2) …..”
What results?
7. How did authors measure sample size?
8. Lines 55-57 have to have some references.
9. There is no clear evidences for measuring research variables.
10. Data were collected in 2016 (based on lines 271-272). Need more explanation how their data were collected.
11. Explain more about questionnaire design. Is there any pilot study?
12. Please prepare reliability and validity of your questionnaire.
13. I would like to see about their analysis regarding their missing data and outliers.
14. It’s better consider a Table to show the results of their hypotheses.
15. Why authors didn’t use Structural Equation Modeling (SEM), or PLS?
16. Figure 1. How do you measure moderating of female? Even for moderating analysis, I haven’t seen any proper data analysis.
17. Table 3 R-square are so low, how the authors can trust the outputs of this modelling.
18. Contribution of the study is not strong enough.
Author Response
Thank you for your valuable advice, and we responded all comments in the attachment.

Round 2
Reviewer 2 Report
The authors well amended the manuscript based on all of my comments. I'm glad to inform you that the paper has enough quality to publish. Thank you and I wish all the best for the authors.